# Four-Dimensional Stimuli-Responsive Hydrogels Micro-Structured via Femtosecond Laser Additive Manufacturing

**DOI:** 10.3390/mi13010032

**Published:** 2021-12-27

**Authors:** Yufeng Tao, Chengchangfeng Lu, Chunsan Deng, Jing Long, Yunpeng Ren, Zijie Dai, Zhaopeng Tong, Xuejiao Wang, Shuai Meng, Wenguang Zhang, Yinuo Xu, Linlin Zhou

**Affiliations:** 1Institute of Micro-Nano Optoelectronics and Terahertz Technology, Jiangsu University, Zhenjiang 212013, China; renyp@ujs.edu.cn (Y.R.); Tongzp@ujs.edu.cn (Z.T.); wangxj0122@hnu.edu.cn (X.W.); 2212003077@stmail.ujs.edu.cn (S.M.); 2Wuhan National Laboratory for Optoelectronics, Huazhong University of Science and Technology, Wuhan 430074, China; chunsan.deng@foxmail.com (C.D.); D201780704@hust.edu.cn (J.L.); zhangwg@hust.edu.cn (W.Z.); xvyinuo@hust.edu.cn (Y.X.); m201772798@hust.edu.cn (L.Z.); 3Whiting School of Engineering, Johns Hopkins University, Baltimore, MD 21218, USA; c9lu@ucsd.edu

**Keywords:** femtosecond laser, additive manufacturing, hyaluronic acid methacryloyl, polyethylene glycol diacrylate, stimuli-responsiveness

## Abstract

Rapid fabricating and harnessing stimuli-responsive behaviors of microscale bio-compatible hydrogels are of great interest to the emerging micro-mechanics, drug delivery, artificial scaffolds, nano-robotics, and lab chips. Herein, we demonstrate a novel femtosecond laser additive manufacturing process with smart materials for soft interactive hydrogel micro-machines. Bio-compatible hyaluronic acid methacryloyl was polymerized with hydrophilic diacrylate into an absorbent hydrogel matrix under a tight topological control through a 532 nm green femtosecond laser beam. The proposed hetero-scanning strategy modifies the hierarchical polymeric degrees inside the hydrogel matrix, leading to a controllable surface tension mismatch. Strikingly, these programmable stimuli-responsive matrices mechanized hydrogels into robotic applications at the micro/nanoscale (<300 × 300 × 100 μm^3^). Reverse high-freedom shape mutations of diversified microstructures were created from simple initial shapes and identified without evident fatigue. We further confirmed the biocompatibility, cell adhesion, and tunable mechanics of the as-prepared hydrogels. Benefiting from the high-efficiency two-photon polymerization (TPP), nanometer feature size (<200 nm), and flexible digitalized modeling technique, many more micro/nanoscale hydrogel robots or machines have become obtainable in respect of future interdisciplinary applications.

## 1. Introduction

The development of modern robotics/sensors [1], micro/nanomechanics [2], tissue engineering [3], and drug delivery [4] introduces an urgent demand for smart micro- or nanostructured machines or robots [5]. However, conventional 3D-printed constructs have fallen short of expectations, mainly due to their bulky volume and inability to mimic the dynamic human tissues. Thereby, the shape-reconfigurable hydrogels emerge as a new scientific frontier for cancer treatment [6], wound healing [7], or biomimetic applications [8,9,10,11,12,13,14,15] acting similar to artificial muscles [11], grippers [12], actuators, active origami or machines [13], swellable scaffolds [14], organic electronics [15], microneedles [16,17], etc.

These smart hydrogel devices would ideally possess the embeddable volume and the reconfigurable morphology to be applied [9,10]. The composition of smart hydrogels re-defines the environment-to-hydrogel interactions, the controllability of responsive behavior [13], and the dynamic mechanical properties to a large extent. In the context of biomimetic applications, hydrogel-based materials have showcased certain advantages such as softness [14], biocompatibility and degradability [18], and massive potential for cell adhesion and proliferation.

Many additive manufacturing methods have been demonstrated with the four-dimensional (4D) time-dependent shape reconfiguration to obtain controllable stimuli-responsive behavior [19]. Four-dimensional printing has revolutionized traditional three-dimensional printing manufacturing products worldwide. In literature reviews, the state-of-the-artwork of 4D printing generally integrates the time-dependent behavior of stimulus-responsive materials [20], the associated materials interacting with various stimuli (physical, chemical, and even biological signals) for pre-designed motion or actuation [21,22]. The already-known 4D examples can be found from two-photon stereolithography [23] to 3D printing [24,25], extrusion fabrication, or ink-writing methods [26,27].

Among the existing macroscopic 4D products and fabrications with a relative millimeter or sub-millimeter resolution, the TPP using the femtosecond laser as a light source captures roaring attention. As is known, the primary-stage products of TPP are mostly stationary without controllable stimuli-inspired properties or artificial shape morphing. We recently reported a carbon nanotube-doped hydrogel with swelling-to-shrinkage behavior using two-photon polymerization (TPP) [15], where the swellable hydrogel scaffold absorbs functional materials for semiconductor applications. Its confinement of nonlinear two-photon absorption (TPA) within the submicron focal volume provided an ultra-fine spatial resolution [28]. Flexible laser parameters and a computer-aided technique [29] promised a tunable formation quality in fabricating quasi-arbitrary three-dimensional (3D) devices. In a nutshell, TPP will be a predominant tool [14,23,28] for fast fabricating core devices in interdisciplinary research due to its structural diversity and selective resolution.

Micro/nanostructures fabricated via TPP are formed through high-density covalent bonding networks. The solidified structures, generally, cannot undergo large transformations such as soft/elastomer materials. Recently, the smart properties of hydrogels have been improved. For example, the traditional actuation uses a residual stress-driven method for thermal shape deformation [30,31]. Some researchers use a unique laser writing process for pre-designed shape morphing [32], or a doping carbon nanotube to enhance the light-responsive behavior [33,34], or use a bio-environment to tune the optical properties [35,36]. Although having achieved tremendous progress [29,30,31,32,33,34,35,36], the stimuli-responsive behavior still deserves further investigation for complex high-freedom shape reconfiguration. The material limitation and monotonous laser scanning methods are now restraining this micro/nanoscale additive manufacturing from innovative development and broadband applications.

Herein, to obtain direct fabricating and temporally controlling micro/nanoscale quasi-arbitrary 3D geometries without filling other functional particles, we tentatively perform the modified TPP (Figure 1) on the bio-compatible hyaluronic acid methacryloyl (HAMA, Figure 1a) [7,17,23] with polyethylene glycol diacrylate [37] for responsiveness (Figure 1a). The shifted laser-scanning space creates a hetero-distribution of polymeric degrees through the formed hierarchical micro/nanostructures (Figure 1c), leading to a controllable surface tension mismatch. Following the programmable shape morphing, we display and identify several reconfigurable micro-scale structures by applying external stimuli for the first time. Shown by the cell loading experiment (Figure 1c and Appendix A), the hydrogels allow fibril cells to crawl on the surface freely after one week, and the summarized cell viability exceeds 98%, implying the as-prepared hydrogels to be ready for cell culture [6,38] or tissue engineering.

Interestingly, the optical double-frequency technique is adopted here for generating the 532 nm green femtosecond laser beam during TPP fabrication for the first time. The double-frequency crystal is inserted in the path to decrease the optical power and wavelength simultaneously. We also use a micro-mechanics platform to characterize the mechanical behaviors of the TPP-fabricated hydrogels. A surface tension analysis expounds the mechanics’ theoretical rationales during the controllable shape morphing.

## 2. Materials and Methods

### 2.1. Material Preparation

Hyaluronic acid methacryloyl (HAMA, 30 wt%), 2-hydroxy-2-methylpropiophenone (2 wt%, molecular structure seen in Appendix A) [39,40], acrylamide (10%, Figure 1c), and poly(2-ethyl-2-oxazoline) diacrylates (PEG-da 475, 55 wt%) were mixed in phosphate-buffered saline (PBS, 4.8 wt%) for the responsive photoresist. Then, the mixture was pre-processed under 30 min ultra-sonication for dispersion and then was magnetically stirred at 800 rpm for 8 h. We purchased HAMA from Aladdin (Shanghai, China), and PBS from HyClone (Logan, UT, USA). All other reagents were purchased from Sigma-Aldrich (St. Louis, MO, USA) and were not purified before usage. The whole fabrication procedure (including development and applying external stimuli) was carried out without light illumination. In the following experimental section, a family of the photoresists in different weight ratios of two monomers were prepared in the same procedure.

### 2.2. Laser System and TPP Additive Manufacturing

A barium metaborate crystal (Ba(BO_2_)_2_) transformed the 1064 nm wavelength femtosecond laser beam from a Ti:Sapphire femtosecond laser (Chameleon-Discovery, Coherent, CA, USA) into a green 532 nm beam (Figure 2) based on the double-frequency effect. Optical power deceased to mW level in the optical path. An expander and an acoustic-optics modulator (AOM), a half-wave plate, a Glan mirror, and an aperture slot were placed in the optical path following our previous femtosecond laser direct writing system [41] (Figure 2a). The terminal biological microscope (IX83, Olympus, Tokyo, Japan) contained a two-dimensional nanometer-step moving platform, in situ charge-coupled device (CCD), and dichroic mirror, where three 20×, 40×, and 100× oil-immersed objectives were optional for different focusing lengths. The normal data slicing technique processed digitalized models such as 3D additive manufacturing [42,43]. The linearly polarized, sequential, near-infrared laser pulse propagated along the bottom-to-top path into the photon-sensitive photoresist without masks for 3D fabrication [44].

During the TPP process [29,44], we carefully adjusted laser power for resonant two-photon absorption ranging from 1.4 to 5 mW to avoid carbonization or incomplete photo-polymerization. The repetition rate of the ultrafast pulsed laser beam was 78 MHz with an approximate 100 fs pulse width. Subsequently, we three-dimensionally moved laser focus inside photoresist tightly on the substrate to form solid gelation. The available moving speeds of x- and y-axis were both 50 µm/s following the pre-designed straight or curvilinear path (Seen in the Appendix A). After TPP, the unsolidified photoresist was washed away by rinsing in alcohol (purity > 99%) for over 5 min.

### 2.3. Measurement

Substrates to be observed by Nova Nano SEM field emission electron microscope (SEM) with acceleration voltages of 5 KV were pre-coated with indium tin oxide semiconductor (ITO) film for electric conduction. The SEM software analyzed the size and spatial resolution. The imaging spectrometer (island-320, Teledyne Princeton Instruments, Princeton, NJ, USA) reflected the fluorescence images of hydrogel structures with a high-resolution scientific complementary metal oxide semiconductor (sCMOS, KURO, Teledyne Princeton Instruments, Princeton, NJ, USA). The software of island-320 set the color of fluorescence images. A CCD camera installed on the digitalized inverted microscope (ix83, Olympus, Tokyo, Japan) recorded the dynamic responsiveness. The minimum time interval of each video was a single sec. The external water stimuli were realized by dropping deionized water onto the sample or blowing moisture from a humidifier to investigate the responsive behavior.

The pH variation was realized by adding the diluted hydrochloric acid into in-solution hydrogels (Appendix A). A light stimulus was applied using the same optical system to the fabrication. The heating condition was realized by placing the sample on the thermoelectric cooler with a digital control step size of 0.2 °C at a range from 20 to 40 °C. The solidified samples’ heat/water micro-forces that occurred were measured by FT-MTA2 of FemtoTools (Switzerland) with FT-S10000-TP tungsten probes of 50 × 50 µm^2^ tip radius and 5 nN resolution. Young’s Modulus of the solidified samples was measured using FT-S100-TP of 2 × 2 µm^2^ tip radius and at the same resolution.

## 3. Results and Discussion

### 3.1. Fabricating Hollow 3D Structures with Selected Spatial Resolution

By laser scanning inside the photoresist (Figure 2b), the smart properties of materials provided fundamental responsiveness as beneficial advantages in devices. For example, the hydrogel-based microlenses (Figure 3a) exhibited the ability to change light facula similar to a dynamic focus lens (Appendix A). Seen in Figure 3a, the diameter of the lens changed from about 20 μm to 29 μm, meaning the swelling area in the X–Y plane increased by at least 100% (seen in the Appendix A). Subsequently, we changed the volume of the hydrogels and measured the volume alternation to confirm an approximate swelling ratio of >210% (seen in Appendix A). Moreover, the as-prepared hydrogel inherited the pH-responsive ability such as the previously studied bio-materials [6,45,46]. By slowly changing the pH value to an acidic environment, the in-solution hydrogel further swelled and stretched itself out (seen in Appendix A). With the micro/nanoscale deformable structures, these smart devices promise broadband applications for embedding bio-conditions.

The experimental observation confirmed the desirable mobility of the photoresist and highly effective two-photon absorption at a mild laser power. The photon-induced cross-linking reaction was confined at the submicron voxel (Appendix A). All complex 3D scaffolds self-stood on substrates in the absence of supportive tools. Both the minimum line width and minimum height could reach 150 nm (Appendix A). The experiment generally concluded a suitable scanning speed from 30 to 140 μm/s with an average optical power distributed from 2 to around 20 mW. To check the formation quality for complex hollow structures widely applied for cell proliferation [9], micro machinery [47] or microfluidic [48], a batch of scaffolds consisting of specific tetrahedrons and cubes was firstly fabricated and characterized (Figure 3b,e).

An ultrafine feature size was observable in the CCD or SEM images (tetrahedron in Figure 3b,c, cubes in Figure 3d,e). By adjusting the scanning speed, the equivalent power exposure dose affected the volume of the cross-linking degree a lot. For example, the spatial resolution of Figure 3b scanned at 10 μm/s was better than that of Figure 3c scanned at 1 μm/s. The same selective ability was manifested again in the shape of the cubes. A higher exposure dose of the pulsed laser beam triggered a higher-level polymerization. Based on this factor, the optical parameters could determine the feature size.

Furthermore, the spectrometer-reconstructed fluorescence images of structures (Figure 3c,e) matched well with the SEM images. The highlighted fluorescence implied that the freestanding hollow structures used pure organic hydrogel materials without hard metal. Due to the bio-compatibility, desirable adhesion, and structural complexity of HAMA, and PEG-da [31], the demonstrated scaffolds promised more practical cell applications. The tests of cell viability and adhesion of hydrogels are ongoing, and will be reported on soon.

Additionally, TPP is typically an additive manufacturing process, where voxels stack up every layer, so the resolution of the laser voxels also plays a vital role in precise control. A smaller laser voxel determines a better spatial arrangement in the same exposure dose. By changing the magnifying ability of objectives, for example, N.A. from 1.2 to 1.43, the fine voxel generated an ultrafine resolution. As compared in Figure 3d,e, samples were scanned with N.A. of objectives = 1.43 and 1.2, respectively. The high accuracy implied the use of small voxels and a tight arrangement. All the demonstrated hydrogels here closely followed the design, although not ideally, as some geometric variations resulted from the intrinsic material properties, development, and observation method.

In addition to the selective spatial resolution and biocompatibility, this 532 nm TPP utilized optical parameters and a material ratio to modulate the mechanical properties in a wide range. As tested using our previously reported micro-mechanics technique [10], a Young’s modulus of as-prepared hydrogels presented a wide range from KPa to MPa (Figure 4a), covering the general requirements from tissue engineering to mechanics. These tunable mechanics proved that hydrogels work as a structural material and functional material simultaneously.

### 3.2. Humidity and Light-Triggered Reverse Shape Morphing

Traditionally, researchers have often combined soft active hydrogels with hard inert materials in dual-layer designs for actuation in multi-step fabrication. Utilizing the different swelling-to-shrinkage degrees of different materials, subject to the environment, for example, the temperature results in self-folding machines. However, TPP incorporating smart hydrogels enables the macroscopic stationary structure to reach a micron-to-nanometer level 4D function using single materials in a single step. The molecular interactive force between the functional groups and applied stimuli (polar solvent, water, acid, or alkali solvent) contributed to the stimuli responsiveness. Therefore, we changed the optical power and spacing width formed during scanning to display the resilient shape deformation ability.

In nature, many plants use water sorption and desorption for motion or reversible shape morphing. To mimic this behavior at a microscale, we fabricated the water-swelling hydrogel. The polymeric matrix consisted of permanent covalent carbon bonds in polymer materials, and various chemical functional groups could collaborate with outside-applied stimuli for judicious motion. As seen in Figure 4b,c, the humidity (or water) reversed the single-layer planar-like hydrogel, and the initial plane changed into a C type. No matter how frequently it is immersed into water or heated, the basic frame of the sample stayed unchanged, which denoted the cross-linked network’s existence as a skeleton for structural integrity.

Here, the water molecules worked as the triggering condition, and the molecular force captured water to swell or shrink by heating to recover. The volume ratio shrank over 200% in evaporation, demonstrating a high water retention (Appendix A). The reproducible volume changing meant that incredibly soft materials with a high liquid content are applicable to various biological and clinical research areas, from osteoporosis through tissue regeneration to hemorrhage control.

Figure 4d illustrates another kind of actuating method, where the hydrogel absorbed light energy and caused a local shape deformation, causing the spider-shaped hydrogel to activate, corresponding to the applied laser beam. In the light-fueled reconfiguration, the formed matrix absorbed photon energy and converted it into mechanical properties. The amplitude, location, frequency, and speed of the shape-changing properties passively depended on the applied laser beam (seen in Appendix A). Here, both the laser pressure [49] and osmotic pressure in the water [50] contributed to the local shape morphing. Without the osmotic pressure (we evaporated the water off), the responsive activity of the spider hydrogel decreased significantly (seen in Appendix A).

Furthermore, we fabricated the tadpole-shaped hydrogel, which swung its tail shape using the swelling effect (seen in Appendix A), where the in-plane tail bent in air but straighten in water (Figure 4e). As a typical reverse process, we could prove the shrinkage of the tail to the initial state (seen in Appendix A). In the discussion, the critical factor, besides the material affecting the bending and stretching, was the groove depth in the tail, which has previously been explored as a mechanism for shape deformation using a self-folding theory based on Timoshenko’s theory [51].

Then, we fabricated a smart two-layer structure (Figure 5a), and the swelling happened out-plane in a perpendicular direction. The planar hydrogel bent upward reversely, and a part of the hydrogel relocated on the substrate due to intrinsic adhesion. The trick for the reconfigurable two-layer structure was the uneven scanning space of two layers. Therefore, the densities of two layer (seen in the SEM image of Figure 5a) varied a lot, leading to an uneven swelling or shrinkage degree and inducing shape morphing on the upper layer. The interface between the two layers was linked by smooth covalent bonding, with no mismatch of the traditional dual-layer design for actuation. Notably, the bending direction was perpendicular to the substrate (Figure 5b), implying a direction control using a two-layer structure. No fracture or physical damage was found in any of the shape-morphing hygromorphic hydrogels. Reverse programmability also meant that the functional groups were well maintained during and after TPP. The micro-structured hydrogels required only several seconds for shape reconfiguration, outperforming those bulky hydrogels of slow diffusive swelling rates [52,53] due to the micro/nanoscale surface effect, which made them more applicable for various aqueous environments.

### 3.3. Heat-Induced Shrinkage Behavior

Besides the humidity or light stimuli for responsiveness, the heating process also led to a self-bending action similar to an artificial muscle (Figure 6). The unique features found by heating, provide possibilities for sensing or actuation as well. If heated, the water uptaken by hydrogel would evaporate. Subsequently, the created surface tension changed to form a shrinkage-based 3D structure. The interspacing of adjacent nanowires modulated the bending degree. Illustrated by the flower (Figure 6a), heart (Figure 6b), and grid structure (Figure 6c), heat-induced deformation became predictable and useful. The hydrogel detected the temperature shifting in the ambient environment and changed its surface tension in the macroscope view.

The heterostructure consisted of solidified hydrogel nanowires, and the smooth spacing resulted in a divergence in shrinkage behavior. This divergence caused residual stress at the molecular level and caused the inward-direction contractile surface tension to accumulate. As seen in Figure 6a, the heat-transferring process differentiated in the eight petals, causing a disorderly shrinkage. Subsequently, we fabricated a symmetrical heart shape, where the heating process showed an asymmetrical shrinkage (Figure 6b). As seen in Figure 6c, another grid hydrogel self-folded into an out-plane uneven ball through heating. Conclusively, the uneven distribution of the geometry intensified the self-folding character and decreased the responsive time. The underlying mechanism for the controllable shape morphing could be found in the explanation section on surface tension (Appendix A). The mechanics platform (seen in Appendix A) further verified the tunable mechanical properties for reverse shape morphing.

## 4. Conclusions

In this study, we succeeded in developing a composite hydrogel material sensitive to a water/light/heat environment with a 532 nm femtosecond laser TPP. Compared to the mainstream optics/electron beam mask-projected stereolithography, the proposed two-photon polymerization held several advantages. An ultrafine feature size was obtained by staking the nanoscale voxel of the TPP system. The conventional macroscopic signal-triggered patterns or structures were miniaturized to a three-dimensional micron/nanoscale. The nonlinear characteristics of the fabrication processes still offered a sub-micron writing resolution, which is of great interest to micron-robotics, nano-drivers, and wearable sensors. Meanwhile, the stimuli-responsive photoresist contained no metal or alloy to improve biocompatibility. The controllable behaviors of the micro/nanostructures were being fatigue-free, environment-inspired, and quickly responsive, promising broad applications in micron actuators, sensors, micro-robotics, and biomimetic fields.

## Figures and Tables

**Figure 1 micromachines-13-00032-f001:**
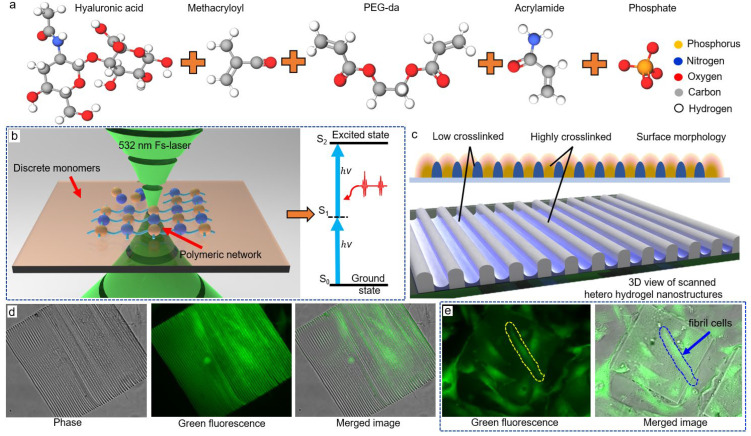
The schematic procedure of TPP incorporating stimuli-responsive hydrogels: (**a**) the main components used in the photoresist containing the methacryloyl-modified hyaluronic acid, PEG-da, and acrylamide as structural or functional materials rendering the hydrogel with stimuli-responsive ability; (**b**) illustration of the laser beam focused on the interface between material and substrate, the laser beam assembles the discrete functional polymers into a polymerized matrix, and the two-photon absorption process; (**c**) description of the used hetero-scanning TPP strategy, where the laser-scanned path forms highly crosslinked nanowires, while the interconnecting spacing is a low-degree crosslinked area, and the 3D view illustrates an interlocking morphology; (**d**) phase image, fluorescence image, and the merged images of typical grating structure created by our proposed method; (**e**) the fluorescence and images of loading fibril cells on the prepared square hydrogels for 2 weeks, where the active cells crawled on the surface of the structure after one week, indicating a desirable adhesion and bio-compatibility for cell culture.

**Figure 2 micromachines-13-00032-f002:**
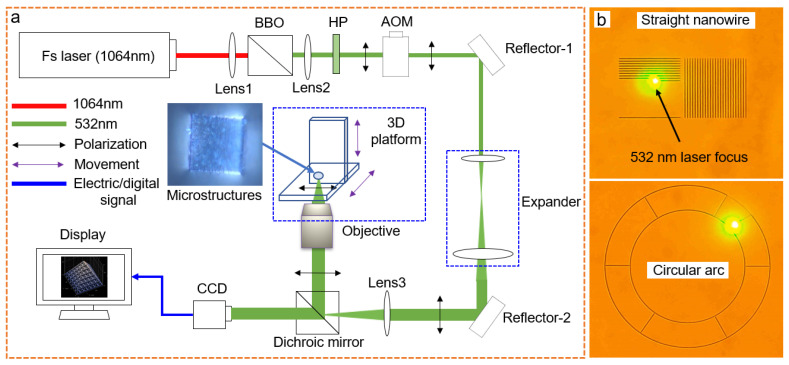
(**a**) The optical configuration of the double-frequency technique in ultrafast laser system for green laser beam; (**b**) the scanned straightforward nanowires and a pattern of curvy lines using the 532 nm femtosecond laser.

**Figure 3 micromachines-13-00032-f003:**
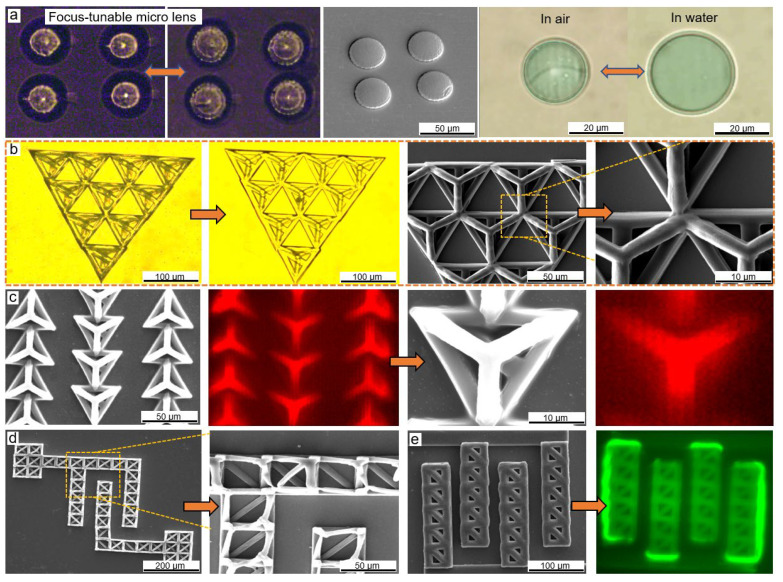
(**a**) A group of dynamic focus lenses using the proposed hydrogel materials, which tuned the size of facula by swelling-to-shrinkage on background light in a dark room; additionally, a comparison of the single hydrogel lens before and after swelling is shown; (**b**–**e**) the fabricated bio-scaffolds of tetrahedrons and cubes, respectively. (**b**) CCD, SEM, and zoomed-in SEM images of a triangle array of tetrahedrons is contained in the first panel; (**c**) SEM and fluorescence images (red) of tetrahedrons scanned at slow scanning speed about 5 μm/s; (**d**,**e**) observation of two arrays of cubes. Zoomed-in view shows a micrometer-level resolution (approximately 2 μm in (**d**), and 16 μm in (**e**)) to self-support the complex cube-stacked structures.

**Figure 4 micromachines-13-00032-f004:**
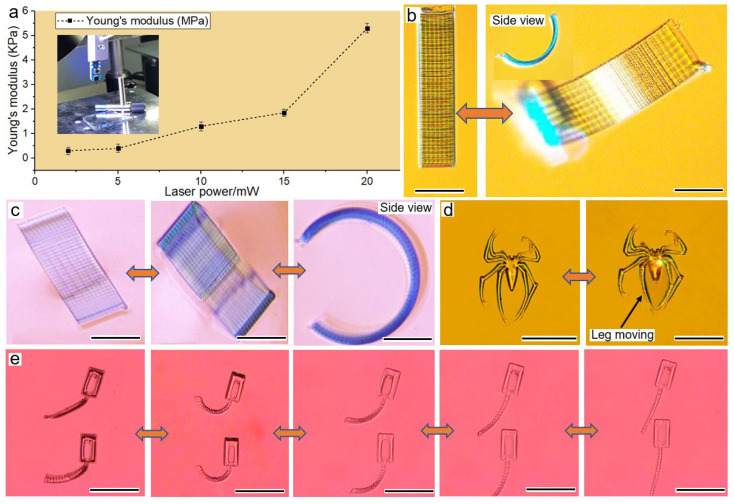
(**a**) Young’s modulus of hydrogels; (**b**) water reversely bent a planar hydrogel from the initial planar to a bent shape, and the scale bar is 50 μm; (**c**) ethanol bent a planar hydrogel to a bent shape as well, and the scale bar is 50 μm; (**d**) a spider-shaped hydrogel was fabricated using our smart materials and then scanned by a laser beam for local actuation, the area projected by laser absorbed photon energy, and thermally swelled to commence shape reconfiguration, and the scale bar is 50 μm; (**e**) the tadpole-shaped hydrogels stretched their tails by swelling, and the scale bar of CCD images is 50 μm.

**Figure 5 micromachines-13-00032-f005:**
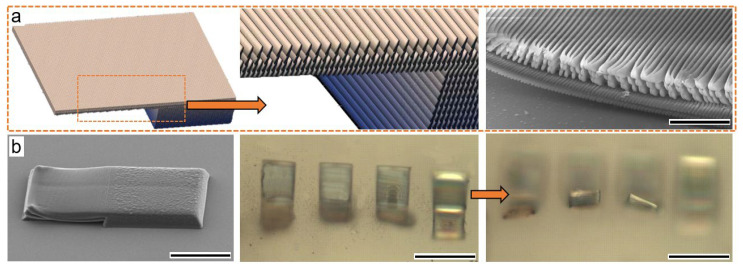
(**a**)The model of dual-layer design, its zoomed-in view, and a side-view SEM image of the differentiated two layers. The upper layer has a higher density of arranged nanowires, and the lower layer has a relatively smooth density; scale bar of the SEM image is 5 μm; (**b**) a two-layer design realizes upward bending out-plane, a group of four two-layer hydrogels demonstrates upward bending observed at height-changed focus. The scale bar of SEM image is 50 μm, scale bar of the CCD image is 100 μm.

**Figure 6 micromachines-13-00032-f006:**
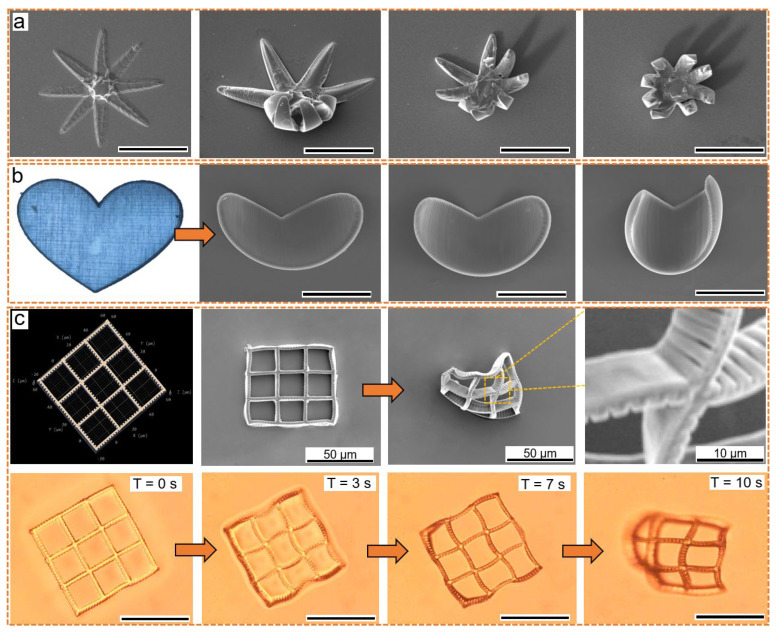
TPP-fabricated thermal-responsive hydrogel for temperature-controlled shaping–morphing: (**a**) the closure of flower mimic hydrogel working as a thermal gripper, where the scale bar is 50 μm; (**b**) an unevenly shrunk hydrogel from initial symmetrical heart shape, where scale bar is 50 μm; (**c**) the model, SEM image, and shrinkage process of a grid-shaped hydrogel, the scale bar is 50 μm.

## Data Availability

Data is contained within the article or Appendix A.

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
