# Peer review of "Four-Dimensional Stimuli-Responsive Hydrogels Micro-Structured via Femtosecond Laser Additive Manufacturing"

_micromachines, 2021, doi:10.3390/mi13010032_

Round 1

Reviewer 1 Report

Micro-structured robots that are stimuli-responsive are a growing topic now for cancer treatments and other drug delivery systems. Here, the authors have described a method to fabricate such microstructured hydrogels and have demonstrated 3D shapes and shapes that are responsive to external stimuli (water/light/heat). The technique to create the hydrogel is already established and well defined (https://pubs.acs.org/doi/pdf/10.1021/acsnano.1c06651). Thus, the novelty of this study would be testing the stimuli responsiveness of the hydrogel. That being said, the authors should consider addressing the following concerns for acceptance.

  1. The biodegradability of the hydrogels during expansion and contraction should be discussed. From lines 217-219, authors have mentioned that the “basic-frame of the samples keeps unchanged”; this has to be quantified experimentally.
  2. In places, authors have discussed the water as an external stimulus; however, quantification of the swelling ratio is missing. Line 221 says over 200% retention, which is not precise. Also, the dimensions of the hydrogel affect the swelling ratio. Hence, a systemic study on dimension vs. swelling ratio and the corresponding time it takes for the response is recommended.
  3. pH variations are majorly tested for the shape-shifting hydrogels for many drug delivery applications. Hence, authors should consider testing their shapes for pH response as well.
  4. Materials and methods: Lines 97-99 can be cut short.
  5. Authors should consider inserting the figure numbers in more places where it is appropriate. Example: Line 169: after “well with SEM images.” Figure number should be mentioned.
  6. Line 171: Is it MAHA or HAMA? Please check.
  7. Figure 4 a, Does the higher laser power affect the dimension of the hydrogel? If yes, does the hydrogel dimension affect the hydrogel response time to external stimuli? Authors should consider discussing these points.
  8. Figure 4 related Supplementary videos: Although scale bars are in this figure, Scale bars are needed in the video as well.
  9. Figure 5 b: Middle and right images are not clear. Consider presenting them in a side view as well.
  10. In the heating stimuli:
  11. What was the heat difference created to introduce a response from the hydrogel?
  12. If the water evaporated from the hydrogel during heating, was the structure regained after removing the stimuli?
  13. The authors should consider adding a graph on response time vs. external stimuli (water/light/heat/pH?) would be helpful.

Author Response

We sincerely thank Reviewer1 a lot for reading and consideration. We are grateful for Reviewer1’s positive comment on English writing, which motivates us to polish the manuscript further. During the reviewing step, we choose the words more carefully according to current related literature, and we check the spell, punctuation marks, and grammar again for clear description. The erroneous places are correspondingly corrected or re-organized.

The changed parts and emphasis are marked in red for checking the resubmitted version. Another version without the red-color mark of the same content is also updated for editing.

Reviewer 2 Report

My suggestion is to reject the article for the following reasons:

  1. English language is below the standard for publication as a journal article. The manuscript could greatly benefit from thorough editing. Much of the grammar and sentence structure is incorrect making it odd to read and many times difficult to understand.
  2. It is not clear what photo initiator (PI) was used? Authors need to discuss the two photon absorption process with respect to the absorption spectra of PI . They should give a conclusive proof that the TPA occurs with 532 nm laser in their material of interest?
  3. Figure 2 shows an SLM added to the setup. What is the use of SLM in the setup. SLM (I guess spatial light modulator), is not mentioned in the manuscript. This needs to be discussed.
  4. Figure 2 c and d. Structure cannot merely depict that the fabricated structure is waveguide or lens. These structures should be tested with light.
  5. Page 3 Line 115. Ultrafast pulsed laser beam is 78 MHz/s. I guess it is 78 MHz.
  6. Page 5 Line 155 “Both the minimum line width and minimum height 155 can reach 150 nm.”. I would expect the height is at least 2.5 time more than the linewidth. Explain?
  7. Page 6 Line 184 “from 20× to 60×, the fine voxel generates an ultrafine resolution.” Resolution is related to Numerical Aperture of objective not the magnification.
  8. Figure 4a. Explain the inset. Figure 4. There is no clear evidence that the response is reversible. Neither this was observed in video files within the supplementary folder. It is not clear from the video or figure 4d that the actuation is happening. Authors need to provide solid proof that the movement of the structure is not due to pressure exerted by laser focus. Figure 4 e. Adding two-way arrows does not mean that the process is reversible. Authors must provide conclusive evidence.
  9. Authors do not provide any evidence that the thermal actuation can be controlled in a user defined way.

Author Response

Dear Reviewer2:

We feel awfully sorry for the negative impressions by our writing errors or inappropriate descriptions. We are regretful to the nonproficient English language, and we sincerely apologize for all inconvenience of editing language or style. We promise to modify the draft throughout to reduce literal misunderstandings. The entire manuscript is carefully verified and re-organized as the Reviewer2’s recommendations. In a nutshell, we will try to eliminate language errors and update the missed discussion or evidence for clarity and readability. Detailed point-by-point response can be found at attachment.

We respect the Reviewer2’s comments and rejection decision. We will accept the final decision after reviewing. No rebuttal will happen if being rejected. We promise to polish this work following Reviewer2’s constructive suggestions for a better presentation quality, and re-consider submitting this work elsewhere after being rejected.

Reviewer 3 Report

The manuscript “Micro-structured stimuli-responsive hydrogels via femtosecond laser additive manufacturing” reports the fabrication of 3D printed microstructures using a laser additive manufacturing method based on the two-photon polymerization (TPP) technique.  The materials used in this manuscript are biocompatible hydrogels based on hyaluronic acid methacryloyl and hydrophilic diacrylate. Interestingly, these hydrogels have a stimuli-responsive capacity; therefore, it is possible to change their mechanical properties or shape via light or humidity exposure.

In general, the manuscript is well-written, and it has enough novelty to be published on Micromachines, but some minor modifications need to be done before publication.

  1. I think that the introduction section could be improved by including the concept of 4D printing. The technology presented in this manuscript fits with the 4D printing, so reformulating the introduction (and maybe the title) based on this new technology could improve the impact of your research.
  2. A detailed study about printing resolution depending on the magnification of the objective used must be done. Just mentioning it in lines 180-188 is not enough. Maybe a graph with resolution evolution must be attractive, like the one reported in figure 4a for the mechanical properties.
  3. The caption in figure 4 is wrong. There are two (c). Also, the images in figure 4d are not explicative enough.
  4. In lines 244-250 you explain the mechanism of the two-layered sample reconfiguration. Is there any evidence that proves this affirmation? Is there any previous study made by your research group or another that demonstrates the bonding between the two layers?
  5. In general, I think that a brief chemical explanation, with his corresponding references, is necessary to explain the material responsiveness (humidity, light, heat).

Author Response

Dear Reviewer3:

We authors were grateful for the feedback and comments from Reviewer 3. We would like to deeply discuss and share the work progress with research peers. We respect all the questions and comments from the reviewer. Thereby, in the revision, we explain the experiment and results according to the reviewer3’s opinions and point out the changed parts.

Round 2

Reviewer 2 Report

My recommendation is “Accept with minor revision.”

My previous comments:

  1. Manuscript could still greatly benefit from thorough editing.
  2. Please mention the information about photoinitiator in the manuscript. I would more expect the absorption band of PI in around 266 nm (rather than in blue) for two photons absorption to occur using 532 nm.
  3. Page 6 Line 184 “from 20× to 60×, the fine voxel generates an ultrafine resolution.” Resolution is related to Numerical Aperture of objective not the magnification. Please mention the objective type with NA in the manuscript. Resolution is defined by abbe’s diffraction criteria, meaning depends on the NA and wavelength.  

My new comments:

  1. Author mentioned “Among the existing macroscopic 4D products and fabrications with relatively milli-meter or sub-millimeter resolution, the TPP using femtosecond laser as heating source captures roaring attention.” I disagree. Fs laser is not considered as a heating source. It is more a light source.
  2. Define TPP in the first place.
  3. Typo mistake: Micromirror or micro lens in Figure 1.
  4. Please move the section (Line 148-159) to result section. My suggestion is to separate result from method section (if possible even from Figure 1 and Figure 2)
  5. Figure 2. Optical setup. Why is laser focused before dichroic mirror?
  6. Line 149-159 is a result and is mentioned in the method section.
  7. Author mentioned “For example,the hydrogel-based microlens (Figure 2c) exhibit the ability to change light facula similar to a dynamic-focus lens.“

These microlenses like structure is shown to shrink. It is unclear if this changes the light facula.

  1. Line 198-199: Figure numbering is wrong.
  2. Please address inconsistency in the scale bar.
  3. Line 290, TPH or TPP??
  4. My suggestion is to move the section “Explanation on surface tension” to SI.

Author Response

Dear Reivewer2:

We sincerely appreciate your consideration, recommendation, and impressive inquires on our clumsy work. We feel grateful to have the great opportunity to improve the presentation and learn more professional knowledge with your review. We accept all your constructive suggestions and pertinent criticism. We strongly believe the merit of this work will be strengthened, and defects will be further reduced through this review. In line with your comments, the related issues are highlighted, discussed, and modified one by one throughout the draft. The answers are still blue, and the emphasis or changed parts are marked in red color here.

Again, we apologize for missing some mistakes or questions in the 1-st review. We are open to other questions or concerns you may offer in the future, and we will unconditionally make any alterations if necessary. We loyally respect your opinion and will provide any assistance you need.

Corresponding to suggestions in E-mail, the revisions made to the manuscript are marked up using the“Track Changes” function in MS Word version.

My recommendation is “Accept with minor revision.”

We, all authors, are excited to see the hope of being accepted for publication with the excellent platform Micromachines. We are grateful to Reivewer 2’s positive comments. We will further correct all the listed issues for better quality.

My previous comments:

  1. Manuscript could still greatly benefit from thorough editing.
  2. Please mention the information about photoinitiator in the manuscript. I would more expect the absorption band of PI in around 266 nm (rather than in blue) for two photons absorption to occur using 532 nm.
  3. Page 6 Line 184 “from 20× to 60×, the fine voxel generates an ultrafine resolution.” Resolution is related to Numerical Aperture of objective not the magnification. Please mention the objective type with NA in the manuscript. Resolution is defined by abbe’s diffraction criteria, meaning depends on the NA and wavelength.  

We respect your comments and suggestions. In the 1-st revision, we followed the comments and made corresponding modifications. In the 2-nd revision, we will recheck the questions. We updated a more suitable photoinitiator and gave more practical objectives parameters.

In the 2-nd revised version, we added two new references “Yu, H.; Ding H.; Zhang, Q.; Gu, Z.; Gu, M.Three-Dimensional Direct Laser Writing of PEGda Hydrogel Microstructures with Low Threshold Power using a Green Laser Beam. Light: Adv. Manuf. 2021,2,3. and Vinck, E.; Cagnie, B.; Cornelissen, M.; Declercq, H.; Cambier, D. Green light-emitting diode irradiation enhances fibroblast growth impaired by high glucose level. Photomed. Laser Surg. 2005,23(2),167-171.” to introduce the background of the photoinitiator.

We also corrected the laser system setup according to your suggestion in the 2-nd review (question 5). We sincerely thank you for reviewing and providing constructive instructions. 

My new comments:

  1. Author mentioned “Among the existing macroscopic 4D products and fabrications with relatively milli-meter or sub-millimeter resolution, the TPP using femtosecond laser as heating source captures roaring attention.” I disagree. Fs laser is not considered as a heating source. It is more a light source.

We are sorry for the careless choosing words in this sentence. The “heating source” is an indeed inappropriate description. During the TPP, the laser beam provides photons as energy for the radical polymerization reaction. The electrons of the photon initiator or monomer absorb the photon energy and excit for the subsequent chemical reaction. Maybe “the source of photon energy” is more suitable.

All authors believe that your viewpoint is scientifically right. In this work, the fs laser beam triggered the polymerization process, also generated fluorescence like a light source. Correspondingly, the “heating source” is changed to “light source”. We appreciate the in-depth consideration of TPP, and we learn much professional knowledge here.

  1. Define TPP in the first place.

We thank you for this suggestion again! We are sorry for the missed definition of TPP in the Abstract section. Now, in line 24, TPP is replaced by “the high-efficiency two-photon polymerization (TPP)”. In the Introduction part, TPP is defined firstly in line 60.  The two-photon absorption is defined as “TPA” in line 62 now.  

We appreciate you pointing out the missed definition or mistakes! We believe the added description will further reduce the confusion in reading.

  1. Typo mistake: Micromirror or micro lens in Figure 1.

Thanks for pointing out this mistake! We are grateful for your question and suggestion! We changed the micromirrors to microlens in new Figure 3.

  1. Please move the section (Line 148-159) to result section. My suggestion is to separate result from method section (if possible even from Figure 1 and Figure 2).

Wonderful suggestion.  We check the content of Line 148-159, and we are sorry for our disorder in the paragraphs. As suggested, the section (Line 148-159) displays our prepared devices (swellable lens), a typical prepared device to prove the feasibility of our laser fabrication method, which should be a part of the result section.

We are deeply sorry for our wrongly placing the section. Figures 2 are now re-produced following your suggestion. The part of the microlens is moved to Figure 3 as result.

  1. Figure 2. Optical setup. Why is laser focused before dichroic mirror?

Great question! It is our mistake, the laser beam travels through a focusing lens before entering into the dichroic mirror. The missed focusing lens3 is added and labeled as below. Figure 2 is re-newed again with labels of three lenses now.

  1. Line 149-159 is a result and is mentioned in the method section.

Many, many thanks for highlighting this error. We are grateful for reviewing this question before the final paper is accepted. This question is answered as the former question 4. As a solution, line 149-159 is moved to the result section with a reproduction of Figures 2 and 3. 

  1. Author mentioned “For example,the hydrogel-based microlens (Figure 2c) exhibit the ability to change light facula similar to a dynamic-focus lens.“

These microlenses like structure is shown to shrink. It is unclear if this changes the light facula.

We are sorry for the vagueness that stemmed from our experimental condition. The experimental observation is conducted in a dark room. Therefore, the presented figure is vague about the swelling process. Our operation order is adding solvent into the lens, not evaporating water from them. Here, we added another video for your observation. In practice, we lack a usable light source in the setup. The experimental light projection is a background light from the digital microscope Olympus IX83 with limited brightness. To observe the facula, we have to close the daylight lamp in the laboratory environment. 

In the newly-uploaded video evidence, the facula is formed by focusing the light. After absorbing the water, the hydrogel lens swelled, not shrunk. Therefore, the tuning of the light facula has a dependence on the swelling ratio. The newly-added supplementary video S2 is added to describe the swelling process.

  1. Line 198-199: Figure numbering is wrong.

The Figure 3 numbering is updated due to the reproduction of Figure 3. The sentence is rewritten as “spectrometer reconstructs fluorescence images of structures (Figures 3c and 3e)”.

  1. Please address inconsistency in the scale bar.

Thanks a lot for the requirement. We rechecked the scale bars and slightly changed the scale bars in Figures 3, 4, and 6.

  1. Line 290, TPH or TPP??

We are sorry for this mistake. It should be TPP, not TPH.

It is corrected now.

Your question is highly appreciated.

  1. My suggestion is to move the section “Explanation on surface tension” to SI.

Good suggestion! In the 2-nd revised version, the section is now moved to SI and cited in the main text.

In the last, please accept our thanking you again for all your work and kind instruction. We are open to all your questions or suggestions you may offer in the near future. We feel lucky to discuss with you in the two-round reviews, which improve our knowledge in a professional view. If asked, we will unconditionally share our progress with you or assist all your work.  

Yours sincerely,

Yufeng Tao,  Jiangsu University

[email protected]  or  [email protected]
